# In transit: Cross-cultural differences in international students' constructively aligned learning experience

**Christian Stamov Roßnagel**◉*

School of Business, Social & Decision Sciences, Constructor University, Bremen, Germany

* cstamovrossnagel@constructor.university

## Abstract

Constructive alignment is widely promoted as a culturally neutral framework for outcome-based higher education, yet most evidence for its benefits comes from Western settings and focuses on positive outcomes. This study examined whether international students' home learning culture moderates the associations between perceived constructive alignment and learning motivation, cognitive load, and academic stress. In a three-wave survey, 129 first-year international students at an English-medium university in Germany reported on one constructively aligned class across a semester. At $T_1$, they rated the clarity of learning outcomes, alignment of teaching activities and assessments, and feedback effectiveness. Motivation (competence self-perception, enjoyment, usefulness) and cognitive load (intrinsic and extraneous) were rated at $T_2$, and academic stress (workload- and expectations-related) at $T_3$. Moderated linear regression analyses showed that higher perceived alignment was linked to stronger motivation in European students, but these benefits were attenuated in students from instructor-centred traditions: the competence benefit was weaker for Gulf Cooperation Council students, and the enjoyment and usefulness benefits were weaker for Confucian Heritage students. Perceived alignment robustly predicted lower extraneous cognitive load across all groups, with no reliable cultural moderation. In contrast, alignment was associated with lower expectations-related stress in European students but with higher workload- and expectations-related stress in Confucian Heritage and Gulf Cooperation Council students. These findings suggest that constructive alignment is not culturally neutral: for students from instructor-centred learning cultures, the same design features that support motivation can simultaneously heighten stress, underscoring the need for culturally responsive implementation in internationalised higher education.

**Data availability statement:** Data are available from a public repository at https://osf.io/vgpyk/.

**Funding:** The author(s) received no specific funding for this work.

**Competing interests:** The authors have declared that no competing interests exist.

## Introduction

The constructive alignment (CA) approach [1] is an internationally established framework for outcome-based higher education [2,3] that has received considerable attention from a policy-making perspective. Seen as valuable for enhancing education through holistic assessment [4] and aligning curricula with intended graduate attributes [5], CA responds to the increasing demand for universities' accountability regarding educational quality and effectiveness [2,6–8]. Consequently, the CA framework has become a quality assurance instrument in several countries and regions, such as Australia [9], the European Union [10], the USA [11], and Vietnam [12] that may support the internationalisation of education [13,14].

A challenge for internationalisation, however, is that culture-specific aspects have received scant attention, even as CA has been adopted as a quality assurance instrument in regions with vastly different pedagogical traditions [15–18]. Constructive alignment has thus come to be seen as a culturally neutral framework, with an implicit assumption that its benefits generalise across learning cultures [3,16,19]. Yet, analysing CA in the Gulf Co-operation Council countries, [3] noted that requiring students to take an active role as the main drivers of learning negated established cultures of learning and conflicted with the tradition of instructor-centred teaching.

The present study is an exploration of this conflict in an international students context where students from instructor-centred learning cultures such as the Gulf Co-Operation Council countries (Bahrain, Kuwait, Oman, Qatar, Saudi Arabia, United Arab Emirates), the Confucian Heritage Cultures (China, Japan, Korea, Singapore, see [20]) or Latin America (see [21,22]) often transition to constructively aligned instruction at their host universities. The broad assumption was that students interpret constructive alignment through the lens of their culture-based attitudes [23–25] and thus experience a cultural mismatch. Acting as a 'competing force', that mismatch may counteract the intended benefits of constructive alignment. The same design features that enhance clarity and motivation for students from student-centred traditions may therefore intensify role insecurity and performance pressure for students socialised in instructor-centred learning cultures (see Table 1).

### Overview of the study

Prior research has demonstrated reliable associations between constructive alignment and a broad range of affective, cognitive, and motivational outcomes, yet most studies have (a) been conducted in Western educational settings, (b) not assessed cross-cultural differences, and (c) focused on positive outcomes, thus not capturing the potentially double-edged constructively aligned learning experience (see Table 2).

The present study seeks to help fill this gap by examining how international students' home learning culture moderates the relationship between perceived constructive alignment at their host institution and a range of positive (motivation, cognitive load) and negative (stress from workload and from expectations) outcomes. Using a three-wave longitudinal design with 129 first-year international students at an English-medium German university—representing European, Confucian Heritage, and Gulf Cooperation Council learning cultures—I tested whether the motivational

**Table 1. Competing forces in the cross-cultural constructive alignment experience.**

| Force 1<br>Intended CA Benefits | ↔ | Force 2<br>Cultural Mismatch |
|---|---|---|
| Clarity of intended learning outcomes | ↔ | Unfamiliar expectations |
| Frequency and types of assessments | ↔ | Heightened performance pressure |
| Active learner role | ↔ | Discomfort with participation |
| Continuous feedback | ↔ | Evaluative surveillance |
| Autonomy support | ↔ | Perceived abandonment by instructor |

**Table 2. Overview of studies examining constructive alignment outcomes.**

| Study | Year | Country | Outcome variables | Culture examined |
|---|---|---|---|---|
| [29] | 2009 | Australia | Learning approaches | No |
| [30] | 2012 | Australia | Academic self-efficacy | No |
| [31] | 2013 | Australia | Satisfaction, assessment clarity, grades | No |
| [32] | 2013 | Australia | Learning experience, clinical relevance | No |
| [33] | 2013 | Hong Kong | Learning approaches | No |
| [34] | 2016 | Australia | Learning outcomes, assessment clarity, pass rates | No |
| [35] | 2018 | Germany | Motivation, perceived pressure, learning outcomes | No |
| [36] | 2020 | Germany | Motivation, learning demands, learning strategies | No |
| [37]. | 2021 | Germany | Learning approaches | No |
| [38] | 2022 | Finland | Learning approaches, outcomes clarity, assessments | No |
| [39] | 2024 | Iran | Writing achievement, learning approach, engagement | No |
| [40] | 2025 | Germany | Intrinsic motivation | No |
| **Present study** | | Germany | Motivation, cognitive load, stress | **Yes** |

and cognitive benefits of CA extend equally across cultural groups, and whether the stress-reducing potential of alignment holds for students from instructor-centred traditions. With curricula and courses designed according to CA principles, that setting allowed for studying the effects of the transition to CA that is typical of many international students' experience. By examining how students' home learning cultures moderate CA's effects, the present study contributes to developing the theoretical boundaries of CA—moving beyond the assumption of universal effectiveness toward a more culturally informed understanding of when and for whom alignment enhances learning. Considering that cultural diversity is growing in classrooms worldwide due to increasing student mobility [26,27], cross-cultural CA research can help ensure quality CA teaching and learning for all students [28].

## CA perceptions and the learning experience

Rooted in the assumption that "what the student does is more important in determining what is learned than what the teacher does" [41], CA is geared towards getting students to construct their knowledge through their learning activities [41,42]. Intended learning outcomes are defined that nominate actions students are supposed to perform. Teaching-learning activities are then aligned with those outcomes so that students can "practise" the nominated actions. Finally, assessments are aligned with both learning outcomes and teaching-learning activities. Taken together, CA seeks to improve learning by keeping students engaged and helping them understand the intended outcome and how learning activities and assessments facilitate achieving those outcomes [43], which can in turn increase student motivation [36,44]. Several studies found positive relationships between CA and outcomes such as increased learning

self-efficacy, higher self-perceptions of subject-matter competence, and more enjoyment of and effort invested into learning [30,31,35–37,39]. Important in the present context, students' CA perceptions can vary widely and be framed by the teaching learning-practices that students are most familiar with. [29] reported that relative to traditional lecture-based courses, some students found learning easier in a novel course that had been re-designed towards clear learning outcomes and more self-directed learning. Seeing the relevance of their active learning motivated them, thus facilitating their learning. Some of their peers, however, perceived the increased self-directed work as less motivating and as "too much of an effort"; getting information "given to you by some lecturer" would be much easier [29]. [45] found that some students viewed outcomes-based education more positively than traditional teaching because of its transparency and clear outcomes that gave them a clearer picture of their learning progress. Other students, however, stated that the transparency helped them abandon 'time-costly' learning for understanding and to use traditional retention-oriented learning strategies to focus only on the required content [45]. These findings imply that international students are likely to perceive CA against their home institutions' teaching-learning practices. A reflection of overarching cultures of learning, those practices are assumed to shape students' expectations and beliefs [24] about instructor and students' roles and behaviours [23,46]. Students from instructor-centred cultures may perceive CA as the opposite of their expectations. In CA, teaching and learning "almost always means something other than talking for an hour while the learner takes notes" [47]. Yet, students are supposed to sit and pay attention to 'receive' knowledge from instructors who lecture to transmit knowledge in instructor-centred learning cultures such as the Gulf region [48–50] or Latin America [22,21,51]. Students from Confucian Heritage cultures are expected to master learning content through memorisation and repetition. They are 'passive learners'; only after mastering the content are students supposed to engage in communication about that content and potentially take a critical stance [20]. Considering that learning cultures are grounded in broad, overarching cultural values [23], forming an essential part of students' socialisation context from primary education onwards [46,52], they can be expected to impact international students' CA learning experience even after their transition to host institutions [53,54]. In the present research, that impact was studied in terms of motivation, cognitive load, and stress.

## CA, learning culture, and motivation

In prior research, students reported that clear learning outcomes, opportunities for self-directed learning, and the alignment of learning outcomes with assessments helped them see the relevance and progress of their learning, and that the active nature of learning motivated them [29,45]. In a quasi-experimental field study [35], learning outcomes emphasised learning for understanding and learning activities were aligned with learning outcomes. However, whilst students in an alignment group took a transfer test, a retention test was announced and administered to students in a misalignment group. The alignment group reported higher competence self-perceptions than the misaligned group [35]. In a survey over one semester [36], differential relationships were found. For instance, high perceptions of learning outcomes clarity were coupled with stronger self-competence perceptions and higher ratings of course usefulness, whilst perceived alignment of learning outcomes with learning activities drove the enjoyment of learning [36]. In sum, perceived CA seems to enhance motivation to the extent that it affords competence experiences and conveys the relevance of learning. However, perceptions of usefulness and competence likely are culturally bound and vary substantially between learning cultures. In Confucian Heritage Cultures countries, for instance, students are supposed to be 'good students' who demonstrate effort to memorise content without challenging that content [20,55]. Gulf region students are expected to place themselves under their instructor's control who would be responsible for transferring knowledge; students in turn would have to willingly and passionately seek that knowledge [3]. Consistently, students in instructor-centred cultures have been reported to depend on instructors, expecting them to provide ready answers [21,22,56–58]. At their host institutions, they may therefore struggle with CA, feeling uncomfortable in an active learner role [59], which may go with doubts about the relevance of learning. Therefore, the first hypothesis was:

**Hypothesis 1.** The positive relationship between perceived CA and motivation in terms of perceived competence, enjoyment, and usefulness of learning is weaker in students from Confucian Heritage Cultures and the Gulf region, relative to European students.

## CA, learning culture, and mental load

Mental workload refers to the relationship between one's mental processing capacity and the perceived amount of processing required by a given task [60], which can be assessed on various dimensions. In a quasi-experimental study [35], learners in the aligned group reported less pressure from learning than the misaligned group. [36] found that higher alignment of learning outcomes with learning activities alignment was coupled with lower temporal demands and the frustration from learning, whilst higher perceptions of feedback effectiveness were associated with lower mental demands (i.e., calculating, deciding, remembering, searching). Some research suggests cultural influences on mental load. In a visual search task, Dutch and Indonesian students attained comparable levels of performance, but Indonesian students rated the task as more demanding. On the other hand, the load measures were more sensitive to changes in task difficulty across trials in the Dutch group [61]. Using objective physiological indicators, [62] investigated the load from questionnaire item wording. Negatively worded items induced higher mental load than positively worded items. This wording effect was more pronounced in non-native English ("L2") speakers. According to cognitive load theory (e.g., [63]), intrinsic load – that is driven by the complexity of the task – was at a similar level between groups, but the Indonesian and L2 learners, respectively, experienced higher extraneous load, which results from the design of learning materials and/or instructions to learners. All participants carried out the same task, intrinsic load therefore was at comparable levels in both groups. Processing negatively worded items required additional processing, relative to positive items, thus increasing extraneous load. In L2 learners, processing in one's second language required even more additional processing, thus reinforcing the negative wording effect. Considering the group differences in cultural values (e.g., individualism, distance), [61] argued the lower measurement sensitivity in the Indonesian group reflected participants' reluctance to report increased workload, which implies they adjusted their ratings, thus increasing extraneous load. Mathematics education research provides indirect evidence of cultural influences. Students from different cultural backgrounds (e.g., China vs. US) are often taught different approaches to the same mathematical problems [64], resulting in skill differences [65] that are assumed to covary with cognitive load. [66] compared Australian and Malaysian students' learning of mathematical problems under three instructional approaches that are associated with different amounts of intrinsic load. Students in the lowest-load group outperformed the other two groups. Between those groups, Malaysians – who had more algebra knowledge – performed better than Australians with an algebraic approach. Due to their stronger foundation in algebra, many Malaysian students integrated, i.e., "skipped" solution steps, whereas no Australian student showed such integration expertise, indicating the algebraic approach imposed higher cognitive load on them [66,67]. In sum, prior research suggests that the cognitive load from the same learning tasks may systematically differ between learners from different cultures, implying that CA might be associated with higher cognitive load if students perceive CA as misaligned with their learning culture. Against this background, the second hypothesis was:

**Hypothesis 2.** The negative relationship between perceived CA and cognitive load is weaker in students from Confucian Heritage Culture and Gulf region students, respectively, relative to European students.

## CA, learning culture, and stress

Defined as the mental distress associated with anticipated academic failure [68] that can result from one's perceived inability to cope with external demands [69], academic stress can hinder learning motivation, thus ultimately affecting academic performance [70]. Conversely, learning motivation and active learning have been linked to lower academic stress levels [71,72]. Therefore, given the positive association of CA with learning motivation [35,36,40], it seems plausible that higher CA perceptions are linked to lower stress perceptions. However, taking an active learner role entails

communication with instructors and peers to clarify assignments and assessments and to contribute to discussions, and can involve challenging peers and professors' positions [25,73]. Such classroom communication has been identified as a stressor to Confucian Heritage Cultures students who have been claimed to be more silent in class, relative to their Western peers [74,75], to be reluctant to participate in classroom activities, and to avoid asking questions [76–80]. In a recent study, active classroom participation was linked to higher levels of stress in Confucian Heritage Cultures students, relative to their non-Asian peers [81]. As a similar relationship seems plausible for students from other instructor-centred cultures, the third hypothesis was:

**Hypothesis 3.** The negative relationship between perceived CA and academic stress is weaker in Confucian Heritage Culture and Gulf region students, respectively, relative to European students.

## Method

### Sample and procedure

All 289 first-year undergraduate international students at an English-medium German university were invited via email on 22$^{nd}$ January 2024, i.e., in the week before the start of the spring semester, for an initial sample of 165 students. The recruitment phase concluded with one invitation reminder sent on 29$^{th}$ January 2024. All students who had completed all three survey waves, and had taken their primary and secondary education in the same country were retained, for a final sample of 129 students (48.8% female, mean age 20.1 years, SD = 1.65, response rate 44.6%), with four groups of students from Confucian Heritage Cultures (CHC, i.e., China, Japan, Korea, Singapore, and Vietnam; n = 39), Gulf Cooperation Council countries (GCC; Bahrain, Kuwait, Oman, Qatar, Saudi Arabia, and the United Arab Emirates; n = 34), and Europe (n = 45). The remaining 11 students were from six countries (Australia, Costa Rica, Ethiopia, Jamaica, Nigeria, USA) that were too diverse to form a coherent subsample, which is thus labelled as "Other Cultures" (OC) in the results section. Participants' majors included Biochemistry, Business Administration, Computer Science, Global Economics and Management, International Relations, Integrated Social Sciences, Mathematics, Sociology, Physics, and Psychology.

The study was conducted in compliance with the WMA Declaration of Helsinki, as well as the standards of the university's Internal Ethics Review Board. Participants were informed in writing that the data would be used in a project on teaching quality. They would share their impressions of course characteristics (e.g., course goals, teaching methods) to study how those contributed to their motivation. There would not be any performance assessments and participation would be voluntary. Participants could receive course credit and could choose to enter a raffle of Amazon® gift vouchers worth 25 € (approx. US$29). To start the survey, participants confirmed their informed consent by ticking the statements a) that they had read and understood the information and b) that they wished to participate. Participants then self-registered in the database used to distribute the three survey waves. They were informed that they could participate anonymously by using an anonymous e-mail address, and that they could request deletion of their survey responses at any point during or after the study. Data were matched across waves with an alphanumeric code participants generated from family members' names and birth dates. Participants reported in the second (T$_1$), seventh (T$_2$), and fourteenth (T$_3$) week on one class of their choice with 14 weekly appointments of 150 min. each. To ensure participants reported on the same class in all waves, they were shown at the start of the second and third waves the title of their chosen class. All questionnaires were used in their original English versions. Some items were adapted to make them class-specific (e.g., "Teachers have unrealistic expectations of me." was changed to "The instructor of this class has unrealistic expectations of me").

### Measures

At T$_1$, participants' demographic data were collected, including major, age, gender, as well as the countries of birth, primary education, and secondary education to ensure participants had been academically socialised in one learning culture. At T$_2$, students rated on five-point Likert-type scales the constructive alignment of their class with the *Constructive Alignment Questionnaire* [82] in terms of learning outcomes clarity ("I had a clear idea of what I was supposed to learn."),

 

alignment of teaching-learning activities ("I was provided a variety of activities that helped me learn what I was supposed to learn."), alignment of assessment tasks ("The assessment tasks addressed what I was supposed to learn."), and feedback effectiveness ("I received feedback that was clear and specific to what I was supposed to learn."). A Constructive Alignment Perception (CAP) score was calculated as the mean of all item responses, with a higher score indicating a higher CA perception. Students also completed 12 items from the *Intrinsic Motivation Inventory* [83] on the subscales perceived competence ("I am pretty skilled at doing the tasks for this class."), enjoyment ("I think this class is very interesting."), and usefulness ("I believe doing the activities in this class is beneficial to me."). Furthermore, students reported with six items from the *Cognitive Load Scale* [84] their levels of intrinsic load ("The topics covered in this class are very complex.") and extraneous load ("The instructions and/or explanations in this class are unclear.") on an 11-point Likert scale. At $T_3$, perceived academic stress was measured on a 5-point Likert scale with six items from the Perceptions of Academic Stress Scale [85] in terms of stress from expectations (e.g., "The instructor of this class is critical of my academic performance") and from classwork ("The workload in this class is excessive."). Higher scores indicate greater perceived academic stress.

## Results

A data screening revealed no violations of regression assumptions, and Cronbach's Alpha was greater than.70 for all scales. Moderated linear regression analyses in SPSS© 29.0 tested whether constructive alignment perception (CAP) predicted motivation, cognitive load, and stress, and whether learning culture (CHC, Europe, GCC, OC) moderated these relationships. Europe served as the reference category; age, major, and gender were controlled but yielded no significant effects.

### CA, learning culture, and motivation

As predicted, CAP was positively associated with all three motivation outcomes—perceived competence, enjoyment, and usefulness—in the reference group. Adding the CAP x culture interaction terms significantly improved model fit for competence ($\Delta R^2 = .069$, $p < .009$) but not for enjoyment or usefulness (see Table 3). The pattern of interactions partially supported Hypothesis 1: the CAP–competence slope was significantly weaker for Gulf region students, whereas the CAP–enjoyment and CAP–usefulness slopes were significantly weaker for Confucian Heritage Cultures students. Thus, the motivational benefits of alignment were attenuated in instructor-centred learning cultures, though the specific dimension affected differed between groups.

### CA, learning culture, and cognitive load

CAP was a robust negative predictor of extraneous load, consistent with the expectation that alignment reduces avoidable processing demands (see Table 4). The interaction terms did not significantly improve model fit, indicating that this benefit

**Table 3. Moderated regression analyses predicting motivation dimensions from constructive alignment perceptions and learning culture.**

| Predictor | Competence B (SE) | Enjoyment B (SE) | Usefulness B (SE) |
|---|---|---|---|
| CAP | .71 (.27)** | .94 (.41)* | .83 (.23)*** |
| CAP × CHC | −.49 (.46) | −1.09 (.59)* | −.67 (.33)* |
| CAP × GCC | −.83 (.39)* | −.16 (.57) | −.25 (.32) |
| R² | .30 | .10 | .29 |

In this and all subsequent tables, *N* = 129. Values are unstandardised regression coefficients (*B*) with standard errors in parentheses from Model 2 (full interaction models). Europe is the reference category; CHC = Confucian Heritage Cultures; GCC = Gulf Cooperation Council countries. Covariates (age, gender, major) were included in all models but are omitted for brevity; *p* < .05; *\*p* < .01; *\*\*p* < .001.

**Table 4. Moderated regression analyses predicting cognitive load from constructive alignment perceptions and learning culture.**

| Predictor | Extraneous load B (SE) | Intrinsic load B (SE) |
|---|---|---|
| CAP | −1.53 (.48)** | .55 (.54) |
| CAP × CHC | .52 (.71) | −0.64 (.91) |
| CAP × GCC | .59 (.69) | −1.42 (.79) |
| R² | .19 | .05 |

generalised across cultural groups. For intrinsic load, neither the main effect of CAP nor the culture interactions reached conventional significance levels. A marginal CAP × GCC interaction (p = .074) was found that should be interpreted with caution due to limited statistical power. Post-hoc power analysis indicates that the current sample size (N = 129) provides approximately 43% power to detect an interaction effect of this magnitude, substantially below the conventional 80% threshold. Detecting such small interaction effects reliably would require substantially larger samples (N ≈ 350–400). Hypothesis 2 was thus partially supported: it held for extraneous load but not for intrinsic load.

## CA, learning culture, and academic stress

The stress findings revealed a reversal pattern that supported Hypothesis 3. For both workload stress and expectations stress, the interaction models explained substantially more variance than main-effects models (ΔR² = .070 and .078, respectively; both p < .01). As Table 5 shows, CAP was associated with lower expectations stress among European students, but the slope reversed direction for students from Confucian Heritage Cultures and the Gulf region: higher perceived alignment coincided with elevated stress in both instructor-centred groups. A similar pattern emerged for workload stress, where CAP showed no relationship for Europeans but was positively associated with stress for students from Confucian Heritage Cultures and the Gulf region. These interaction effects were among the strongest observed in the study.

## Discussion

This study examined how international students' constructive alignment perceptions relate to learning motivation, cognitive load, and academic stress, and whether these associations vary by learning culture. Consistent with the assumption that constructive alignment can enhance student focus and productive effort [1,86], perceived constructive alignment was positively related to motivation—most consistently for perceived usefulness and competence—for European students, whilst significant cross-cultural interactions indicated weaker slopes in the Confucian Heritage Cultures and/or Gulf region groups. As predicted by cognitive load theory, higher alignment perceptions were associated with lower extraneous load, suggesting that constructively aligned courses can clarify tasks and reduce ambiguity, which in turn diminishes extraneous processing demands [87–89]. Moderation by learning culture was limited, however, and intrinsic load showed no reliable main effect of perceived alignment. A reversal effect emerged for stress: perceived alignment was associated with lower expectations-related stress in European students, but higher stress levels in both Confucian Heritage Culture and Gulf

**Table 5. Moderated regression analyses predicting academic stress from constructive alignment perceptions and learning culture.**

| Predictor | Workload stress B (SE) | Expectations stress B (SE) |
|---|---|---|
| CAP | −.22 (.20) | −.39 (.20)* |
| CAP × CHC | .78 (.28)** | 1.03 (.29)*** |
| CAP × GCC | .81 (.28)** | .91 (.29)** |
| R² | .35 | .37 |

region students. In those groups, perceived constructive alignment was also associated with higher workload-related stress, whereas no significant relationship was found in European students.

## Implications for theory

Prior studies of constructive alignment (CA) have consistently documented its positive effects on student motivation, self-efficacy, and learning approaches in Western higher education contexts [29–31,34–37,40]. However, that research—conducted predominantly in Australia, Germany, and Finland—has not examined whether CA's benefits generalise across learning cultures. Extant studies typically treated learning culture as a background or demographic characteristic and focused primarily on positive outcomes such as motivation and learning approaches, implicitly assuming that CA functions as a culturally neutral pedagogical framework. The present study addressed this gap by explicitly conceptualising culture as a potential moderator of the relationship between CA and both motivational ("positive") and stress-related ("negative") outcomes across student groups from European, Confucian Heritage, and Gulf-region learning traditions. Compared with prior CA research, this design makes two key shifts: from treating culture as a descriptive variable to a theory-based moderator, and from focusing predominantly on benefits to systematically examining both beneficial and potentially adverse outcomes. By broadening the range of outcomes (including cognitive and stress-related demands) and systematically varying learning culture, the study moves beyond the assumption of universal CA effects and tests whether alignment operates similarly across diverse educational traditions.

The present findings extend constructive alignment theory by demonstrating that its assumed benefits are culturally contingent rather than universal. For European students, CA operated largely as the theory and prior Western studies would predict. Perceived alignment enhanced motivation and reduced perceived ambiguity in line with self-determination and CA accounts of clarity and autonomy-support that link clear structure and high-quality feedback to satisfaction of competence needs and adaptive engagement [90–92]. In this sense, the current results converge with the bulk of CA research reporting positive effects on motivation, learning approaches, and engagement in Western higher education. For students from instructor-centred learning traditions, however, the pattern diverged from this literature. The same design features that provide structure and transparency for European students—continuous assessment, explicit performance criteria, and an active learner role—were associated with heightened workload and expectation-related stress among Confucian Heritage and Gulf-region students, despite comparable or even elevated motivational benefits. This dual-effect pattern, in which alignment simultaneously supports motivation yet amplifies perceived performance demands, has not been documented in prior CA studies. These results support recent conceptual analyses [3,16,19] positing that CA is not a culturally neutral pedagogical framework but one whose effects are filtered through students' cultural scripts about teaching, learning, and responsibility.

## Implications for practice

The dual-effect nature of constructive alignment calls for culturally responsive implementation to retain the benefits of alignment while mitigating culture-specific stress responses. This might involve, for instance, sequencing low-stakes, scaffolded practice opportunities before higher-stakes tasks to normalise participation and reduce fear of public evaluation, Additionally, providing dialogic feedback and building students' feedback literacy to support uptake should be helpful [91,93]. Professional development can focus on how to communicate the rationale for active, aligned tasks to culturally diverse cohorts and how to balance autonomy support with clear structure [92].

The cross-cultural differences identified here also bear implications for the integration of AI tools into constructively aligned curricula. AI adoption and perceived ease of use have been shown to vary significantly by cultural background, with international students showing both higher adoption rates and stronger effects of perceived ease of use on attitudes toward AI [94]. Given that students from instructor-centred cultures may experience heightened stress under aligned conditions, AI-augmented CA environments should be designed to prevent increased performance

pressure—for example, through continuous automated feedback—and/or to mitigate stress by providing personalised, low-stakes practice opportunities. The cultural biases embedded in AI systems present an additional consideration: generative AI trained predominantly on Western data may reinforce the pedagogical assumptions of student-centred learning cultures, potentially compounding the cultural mismatch experienced by Confucian Heritage Culture and Gulf region students.

## Conclusion

The present results suggest that CA is not a culturally neutral pedagogical framework, as its effects are filtered through students' cultural scripts about teaching and learning. For learners socialised in systems where instructors bear primary responsibility for knowledge transmission, the active and accountable learner role embedded in CA may be experienced as discontinuity—a mismatch between expected and actual pedagogical norms—that converts transparency into pressure. A dual-effect perspective on CA, which recognises that alignment's benefits and costs can coexist and may be differentially weighted across cultural contexts, therefore offers a more accurate and culturally informed account of when and for whom alignment enhances learning.

This study has several limitations. First, it was conducted at a single institution. While this limits generalisability, it does not invalidate the findings; rather, it provided a controlled setting to isolate learning-culture effects, as all students studied within the same institutionally implemented CA framework. Future research could use multi-site designs to reveal the interplay of cultural and institutional influences and allow for finer-grained group comparisons.

Second, whereas our sample of 129 students provided adequate power for detecting the observed main effects and the larger interaction effects for motivation and stress outcomes, it was underpowered for detecting smaller interaction effects, as evidenced by the marginal CAP × GCC interaction for intrinsic cognitive load ($p = .074$, $f^2 \approx .023$). As interaction effects typically require 4–16 times larger samples than main effects, particularly for small effect sizes, future studies should employ substantially larger samples ($N \geq 350–400$) to conclusively test subtle cultural moderation effects. This power limitation does not affect our primary findings regarding motivation and stress, where robust and theoretically coherent interaction patterns emerged, but it does underscore the need for replication of the cognitive load findings with adequately powered designs.

Third, both constructive alignment perceptions and learning outcomes were assessed merely by self-report measures, and the countries of primary and secondary education were used as broad learning-culture proxies. In future studies, more direct measures of prior pedagogical experiences, values, or culturally based learning preferences could be used, and students' learning experience could additionally be captured with qualitative methods (e.g., semi-structured interviews) and/or in a more longitudinal manner (e.g., using experience-sampling methods) to better understand the processes of CA influence on students' learning experience.

Finally, this study does not address the role of artificial intelligence in contemporary learning environments. While data collection predated the integration of generative AI tools into the studied university's learning environment, emerging research suggests that AI may intersect with cultural factors in significant ways relevant to our findings (see [95]). For instance, AI adoption rates and attitudes have been found to differ substantially between Chinese and international students [94], and generative AI tools embed culturally specific assumptions—exhibiting more interdependent cognitive styles when prompted in Chinese versus more independent styles in English [96]. This implies that AI-mediated learning environments may differentially affect students from different cultural backgrounds by amplifying or mitigating the cultural mismatch effects observed here. Furthermore, research on international students' adaptation suggests that generative AI can function as a 'digital adaptive resource' by reducing information asymmetry and supporting cultural rule interpretation [97]. Future research could examine how AI-assisted learning tools can help bridge cultural gaps in constructively aligned environments—for instance, through culturally adaptive scaffolding—without exacerbating inequities by embedding Western pedagogical assumptions into algorithmic assistance [96].

## Acknowledgments

My thanks go to Weijian Huang and an anonymous reviewer for their constructive comments on an earlier version of this paper. Furthermore, I wish to thank Isabella Lange and Kayla Immanea Collett for their assistance with data collection and data analysis.

## Author contributions

**Conceptualization:** Christian Stamov Roßnagel.

**Formal analysis:** Christian Stamov Roßnagel.

**Investigation:** Christian Stamov Roßnagel.

**Methodology:** Christian Stamov Roßnagel.

**Project administration:** Christian Stamov Roßnagel.

**Writing – original draft:** Christian Stamov Roßnagel.

**Writing – review & editing:** Christian Stamov Roßnagel.

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
