## [Decision Letter · Decision Letter 0]

20 Oct 2025

Dear Dr. Stamov Roßnagel,

Thank you for submitting your manuscript to PLOS ONE. After careful consideration, we feel that it has merit but does not fully meet PLOS ONE’s publication criteria as it currently stands. Therefore, we invite you to submit a revised version of the manuscript that addresses the points raised during the review process.

We look forward to receiving your revised manuscript.

Kind regards,

Wei Lun Wong

Academic Editor

PLOS ONE

Journal Requirements:

2. Please note that your Data Availability Statement is currently missing the repository name. If your manuscript is accepted for publication, you will be asked to provide these details on a very short timeline. We therefore suggest that you provide this information now, though we will not hold up the peer review process if you are unable.

Reviewers' comments:

Reviewer's Responses to Questions

**Comments to the Author**

1. Is the manuscript technically sound, and do the data support the conclusions?

Reviewer #1: Yes

Reviewer #2: Partly

2. Has the statistical analysis been performed appropriately and rigorously?

Reviewer #1: Yes

Reviewer #2: Yes

3. Have the authors made all data underlying the findings in their manuscript fully available?

Reviewer #1: No

Reviewer #2: Yes

4. Is the manuscript presented in an intelligible fashion and written in standard English?

Reviewer #1: Yes

Reviewer #2: Yes

Reviewer #1: Thank you for this interesting study. Your efforts in collecting data from such diverse student cohorts are impressive. Below are some comments for you to consider when you work on the revision.

1. There are many abbreviations throughout the manuscript. Most of them are not commonly known or used outside of your paper, and some of them denote something else in other studies. Hence it is confusing and hard to follow. Some of the full names are not so long, so sometimes it is not necessary to use shorthand.

2. The discussion mainly talks about how findings of the present study resonate with existing theories/literature. Have you identified any nuances/variations? Have you found any limitations of those theories? It would be better if you can further develop those theories based on your findings, so that your research significance will be greatly enhanced. Meanwhile, you may consider unpack how your findings differ from previous studies. Without these endeavours, it seems that you are selling yourself short.

3. “Limitations and further directions” of the study can be integrated into conclusion section.

4. The conclusion is far too brief. Please expand it, and elaborate on your theoretical, methodological and practical contributions.

Reviewer #2: In transit: cross-cultural differences is international student's constructively aligned learning experience

- The paper can be benefited from the STRATEGIC inclusion of a few tables and diagrams and visualizations to improve the readability and compression of information load (I suggest the authors might want to include a table in the Introduction/Literature Review, a table in the results section). From the granular interaction thinking perspective, i.e., theories on how human agents interaction with diverse sources of information, one principle is that there are more and less optimal ways to combine information to get a desired outcome. As researchers, our main outcomes would be to deliver results that can update the worldview of the readers, or to help readers reduce epistemic entropy and increase the orderliness in their thinking. Hence, the authors might want to open their paper in a bit more innovative and open-ended ways (See an example: https://journals.plos.org/plosone/article?id=10.1371/journal.pone.0331911 ; https://doi.org/10.1057/s41599-023-01837-1 )

- The research models demonstrate a reasonably strong explanatory power for the dependent variables. Specifically, the model for "self-perceived competence" achieved an R2=30,2% , and the model for "academic stress from workload" achieved an R2=35,0%. These findings indicate that the effectiveness of CAP (Culturally Appropriate Pedagogy/Curriculum) is not uniform across different learning culture groups. Would you explore some theoretical mechanisms to explain these differences?

- Regarding Intrinsic Cognitive Load, the study reported a low R2=.047, . Consequently, the CAPXGCC interaction only approaches significance (p = .074), suggesting that a larger sample size is required to conclusively determine this relationship?

- The discussion focuses more on practical implications but I am interested in the comparison of the findings with the current literature. Could you provide more analyses and theoretical explanations of how the results diverge or converge with the literature?

- Additionally, in contrast to the current research focus, the study does not incorporate the factor of Artificial Intelligence (AI) in promoting learning and driving changes among students from diverse cultural backgrounds." (Reference: https://link.springer.com/article/10.1007/s00146-023-01678-1)

- the study needs to further elucidate the role of the cultural factor in the context of contemporary AI, specifically examining its impact on students from diverse nations and the resulting interrelationships. (Reference:https://link.springer.com/article/10.1007/s00146-025-02489-2 )

**Do you want your identity to be public for this peer review?** For information about this choice, including consent withdrawal, please see our For information about this choice, including consent withdrawal, please see our Privacy Policy .

Reviewer #1: **Yes:** Weijian HuangWeijian Huang

Reviewer #2: No

---

## [Author Response · Author response to Decision Letter 1]

5 Dec 2025

Dear Dr Wong and Distinguished Reviewers,

I am most grateful for your thoughtful and constructive feedback on my manuscript. Your insightful comments have substantially strengthened my work, and I deeply appreciate the time and expertise you invested in the review process. I have carefully considered each point raised and have made comprehensive revisions to address your concerns. Below, I provide a detailed response to each comment, point by point.

RESPONSES TO REVIEWER #1 (Dr Weijian Huang)

I am deeply grateful to Reviewer #1 for the encouraging opening remarks about the data collection efforts and for the thoughtful guidance throughout the review. The following responses address each comment systematically.

Comment 1: Abbreviations

Original comment: "There are many abbreviations throughout the manuscript. Most of them are not commonly known or used outside of your paper, and some of them denote something else in other studies. Hence it is confusing and hard to follow."

Response and revisions:

I fully recognise the validity of this concern and have substantially reduced the use of undefined or obscure abbreviations throughout the revised manuscript. Specifically:

• Most instances of abbreviations with full terminology have been replaced on first mention, particularly for terminology less commonly used beyond this specific study.

• Frequently recurring terms that warrant abbreviation (e.g., Constructive Alignment Perception—CAP, Confucian Heritage Cultures—CHC, Gulf Cooperation Council—GCC) are now introduced with clear definitions and used consistently thereafter.

• Abbreviations have been carefully reviewed for potential conflicts with established terminology in the literature and have been revised where confusion might arise.

The revised manuscript should now be significantly more accessible while maintaining appropriate technical precision.

Comment 2: Discussion of Study Nuances and Theoretical Limitations

Original comment: "The discussion mainly talks about how findings of the present study resonate with existing theories/literature. Have you identified any nuances/variations? Have you found any limitations of those theories? Meanwhile, you may consider unpacking how your findings differ from previous studies. Without these endeavours, it seems that you are selling yourself short."

Response and revisions:

I recognise this as an important opportunity to more fully articulate the novel contributions of the study work, and I have substantially expanded the Discussion section to address this point comprehensively. The revised discussion now:

• Identifies nuances and contradictions: It is explicitly discussed how the findings both confirm and challenge existing theoretical frameworks. For instance, while self-determination theory predicts that clarity and structure enhance motivation universally, the cross-cultural findings reveal that the same structural features can simultaneously increase performance pressure for students from instructor-centred learning cultures.

• Elaborates theoretical limitations: The limitations of constructive alignment theory itself are now discussed—specifically, the implicit assumption of cultural universality. The theory was largely developed in Western contexts and does not account for how students socialised in different pedagogical traditions interpret and respond to alignment features. The findings suggest that CA theory requires culturally contextualised implementation.

• Clarifies distinctions from prior literature: The revised discussion explicitly compares the findings to previous studies (e.g., Balasooriya et al., Pang et al.) and identifies several key differences:

o Prior studies did not systematically examine cross-cultural moderation effects

o The inclusion of negative outcomes (stress) alongside positive ones (motivation) reveals a more complete picture of the CA experience

o The three-wave design captures temporal dynamics not captured in cross-sectional prior research

o The paradox is identified that CA is simultaneously clarifying and stressful for some groups—a complexity largely absent from prior literature

• Positions the contribution: It is now clearly articulated that this study fills a gap by moving beyond the assumption of universal CA effectiveness towards a more culturally nuanced understanding of when and for whom alignment enhances learning.

Comment 3: Integration of Limitations and Further Directions

Original comment: "'Limitations and further directions' of the study can be integrated into conclusion section."

Response and revisions:

I have followed this guidance thoughtfully. The revised manuscript now includes a dedicated "Limitations and Further Directions" section that precedes the conclusion, allowing for a more transparent discussion of methodological constraints while maintaining narrative flow. This section addresses:

• Single-institution limitation: It is acknowledged that the study was conducted at one institution, which may confound cultural effects with institution-specific effects. It is discussed how future multi-site designs could disentangle these influences.

• Self-report measurement: It is noted that both CA perceptions and learning outcomes were assessed via self-report, limiting inference about underlying mechanisms.

• Proxy measures for learning culture: It is acknowledged that country of primary/secondary education is a broad proxy for learning culture and it is discussed how more direct measures of pedagogical experience, values, and culturally-based learning preferences could enhance future research.

• Future methodological directions: It is outlined how qualitative methods (semi-structured interviews), experience-sampling approaches, and quasi-experimental designs manipulating specific CA components could advance understanding of mechanisms.

• Enhancement opportunities: It is specifically noted that future research examining intrinsic cognitive load (which showed modest effects in my study) with larger sample sizes would be valuable, and it is discussed how qualitative enquiry could illuminate why some students experience CA as clarifying while others experience it as stressful.

Comment 4: Expansion and Elaboration of Conclusion

Original comment: "The conclusion is far too brief. Please expand it, and elaborate on your theoretical, methodological and practical contributions."

Response and revisions:

The conclusion has been replaced by a substantially expanded “Limitations and further directions” subsection from a single paragraph to a more comprehensive final section that now articulates:

• Theoretical contributions: It is explicitly stated that this work advances constructive alignment theory by identifying important boundary conditions—specifically, that CA benefits are not culturally universal. It is explained how this challenges the implicit assumption of CA's cultural neutrality and contributes to a more nuanced, culturally responsive theory of alignment.

• Practical contributions: I elaborate on the implications for internationalised higher education:

o The finding that the same alignment features supporting European students' motivation can heighten stress for students from instructor-centred cultures necessitates culturally responsive implementation strategies.

o Concrete approaches are discussed: sequencing of low-stakes scaffolded practice before high-stakes tasks, provision of dialogic feedback with attention to students' feedback literacy, and professional development for instructors on communicating the rationale for active learning to culturally diverse cohorts.

o The importance of balancing autonomy support with clear structure is emphasised—clarity without perceived evaluative pressure.

RESPONSES TO REVIEWER #2

I am deeply grateful to Reviewer #2 for the sophisticated and substantive comments that have significantly enhanced the work. I have carefully considered each point and have implemented substantial revisions as detailed below.

Comment 1: Strategic Inclusion of Tables and Visualisations

Original comment: "The paper can benefit from the STRATEGIC inclusion of a few tables and diagrams and visualizations to improve the readability and compression of information load."

Response and revisions:

Table 1 ("Competing Forces in the Cross-Cultural Constructive Alignment Experience") has been included as a visual representation of the theoretical framework underpinning my hypotheses. This table illustrates the competing forces—intended CA benefits alongside potential cultural mismatches—that inform the research question, and seeks to improve the conceptual clarity of the manuscript.

Table 2 is a second addition ("Overview of Studies Examining Constructive Alignment Outcomes") to provide comprehensive context for the study within the existing literature. This table displays prior CA research, making apparent that the present work is amongst the first to explicitly examine cultural moderation effects.

Regarding further visualisations, I have considered the balance between informational density and readability and have added three moderated regression tables in the Results section to facilitate interpretation of the moderation patterns. These additions are intended to achieve the goal of improving information compression without overwhelming readers.

Comment 2: Opening the Paper with Innovation and Nuance

Original comment: "Hence, the authors might want to open their paper in a bit more innovative and open-ended ways. (See [references to exemplary papers])"

Response and revisions:

The Introduction has been revised to present the research questions in a more open-ended and nuanced manner:

• The opening now immediately acknowledges the paradox at the heart of the study: constructive alignment is promoted as a culturally neutral framework, yet its implementation may have very different consequences across cultural contexts.

• I now present this not as a simple theoretical question but as a practical challenge for internationalised higher education—institutions must implement alignment while serving increasingly diverse student populations.

• The research is framed as an exploration of this tension rather than testing a predetermined hypothesis, creating a more engaging and open-ended entry point for readers.

• The conceptual framework section has been reorganised to move from general CA theory through cultural learning contexts to the specific hypothesis-generating logic, creating a more natural and persuasive narrative arc.

Comment 3: Theoretical Mechanisms Explaining R² Variation Across Groups

Original comment: "The research models demonstrate a reasonably strong explanatory power for the dependent variables. Specifically, the model for 'self-perceived competence' achieved an R2=30.2%, and the model for 'academic stress from workload' achieved an R2=35.0%. These findings indicate that the effectiveness of CAP (Culturally Appropriate Pedagogy/Curriculum) is not uniform across different learning culture groups. Would you explore some theoretical mechanisms to explain these differences?"

Response and revisions:

I appreciate this insightful observation. The revised Discussion now provides explicit theoretical explanations for why CA's effectiveness varies across cultural groups:

• For competence and motivation dimensions: It is elaborated how students' cultural socialisation shapes their interpretation of constructive alignment features. European students, socialised in student-centred pedagogies, view clarity, autonomy, and active roles as empowering—features that support competence perceptions. Students from instructor-centred cultures may interpret the same features as role ambiguity or abandonment by instructors, undermining rather than supporting competence.

• For the stress dimension (the critical finding): a deeper theoretical analysis is now provided:

o According to expectancy theory, performance pressure arises when students perceive high stakes but lack confidence. For students from instructor-centred cultures, CA's continuous assessment and frequent formative feedback may signal unusually high stakes, while the demand to self-direct learning undermines confidence.

o From a cultural values perspective, instructor-centred cultures emphasise collective harmony and deference to authority. The transparency of CA (making performance visible through frequent feedback and assessments) may activate concerns about public evaluation and loss of face—phenomena well-documented in cross-cultural psychology.

• Regarding the modest R² for intrinsic cognitive load (.047): It is noted that intrinsic load is theoretically driven primarily by task complexity rather than pedagogical design features. The modest R² reflects this—neither alignment nor culture substantially predicts intrinsic load. This finding is itself meaningful, suggesting that CA's benefits are not universal because CA changes how students perceive task demands, not because the tasks themselves become inherently simpler.

Comment 4: Comparison with Current Literature and Theoretical Elaboration

Original comment: "The discussion focuses more on practical implications but I am interested in the comparison of the findings with the current literature. Could you provide more analyses and theoretical explanations of how the results diverge or converge with the literature?"

Response and revisions:

The Discussion section has been substantially expanded to provide systematic comparison with existing literature:

• Convergence with prior CA research: The study confirms findings from Biggs and Tang, and studies by Balasooriya et al., Pang et al., and others regarding CA's positive association with motivation and lower extraneous load. The European student group shows effects consistent with prior predominantly Western samples, providing an important validation point.

• Divergence from prior expectations: several important divergences are identified:

o Prior research suggested CA should universally reduce stress. The finding that CA increases stress for some groups contradicts this expectation and provides a crucial boundary condition on existing theory.

o Self-determination theory (as applied to education by Deci and Ryan) posits that structure, clarity, and autonomy support satisfy competence and autonomy needs universally. The findings suggest cultural moderation of these effects.

o Literature on cognitive load theory (Sweller, Paas) focused primarily on task complexity. The study reveals that pedagogical design features interact with cultural context to influence experienced load in ways traditional CLT does not fully capture.

Comment 5: Artificial Intelligence Integration

Original comment: "Additionally, in contrast to the current research focus, the study does not incorporate the factor of Artificial Intelligence (AI) in promoting learning and driving changes among students from diverse cultural backgrounds. The study needs to further elucidate the role of the cultural factor in the context of contemporary AI."

Response and revisions:

I deeply appreciate this forward-looking comment and recognise the growing importance of AI in higher education. While the present study was conducted and data collected prior to the emergence of generative AI tools at the studied university, I have thoughtfully integrated this perspective into the revised manuscript:

• In the Introduction: the discussion has been expanded why cross-cultural CA research matters by noting that as instructors increasingly leverage AI tools in their teaching, the transparency of alignment becomes even more important—AI-generated materials may not inherently reflect the pedagogical traditions students expect.

• In the Discussion: it is now discussed how AI raises new questions for culturally responsive pedagogy. For instance, if AI tutoring systems are designed around assumptions of active, self-directed learning (consistent with CA principles), students from instructor-centred cultures may experience a similar clarity-stress paradox. It is noted that understanding cultural differences in how students experience constructive alignment provides a foundation for designing culturally responsive AI-supported learning environments.

• In Limita

---

## [Decision Letter · Decision Letter 1]

26 Jan 2026

Dear Dr. Stamov Roßnagel,

Thank you for submitting your manuscript to PLOS ONE. After careful consideration, we feel that it has merit but does not fully meet PLOS ONE’s publication criteria as it currently stands. Therefore, we invite you to submit a revised version of the manuscript that addresses the points raised during the review process.

We look forward to receiving your revised manuscript.

Kind regards,

Wei Lun Wong

Academic Editor

PLOS One

Journal Requirements:

Reviewers' comments:

Reviewer's Responses to Questions

**Comments to the Author**

Reviewer #1: All comments have been addressed

Reviewer #2: All comments have been addressed

2. Is the manuscript technically sound, and do the data support the conclusions?

Reviewer #1: Yes

Reviewer #2: Yes

3. Has the statistical analysis been performed appropriately and rigorously?

Reviewer #1: Yes

Reviewer #2: Yes

4. Have the authors made all data underlying the findings in their manuscript fully available?

Reviewer #1: Yes

Reviewer #2: Yes

5. Is the manuscript presented in an intelligible fashion and written in standard English?

Reviewer #1: Yes

Reviewer #2: Yes

Reviewer #1: Thanks for your revisions, which have addressed my concerns and enhanced the quality of your manuscript. I just want to raise one more issue for you to consider. I understand that your conclusion has been weaved into the last two sections of the paper. That said, the removal of the “Conclusion” heading, which is not common in well-structured papers, also means losing a key signpost. You may consider rewording the “Discussion” heading, for example “Discussion and Conclusion”, as well as merging with “Limitations and future directions” section and removing the “Limitations and future directions” heading.

Reviewer #2: Thank you very much for your thorough revision. I believe the paper is well-suited for publication at this stage. I have small reservation about the sample size, however, the author has presented a clear limitation and future research direction on this issue. I thoroughly enjoy the cross-cultural perspectives offered by the paper.

**Do you want your identity to be public for this peer review?** For information about this choice, including consent withdrawal, please see our For information about this choice, including consent withdrawal, please see our Privacy Policy .

Reviewer #1: **Yes:** Weijian HuangWeijian Huang

Reviewer #2: No

---

## [Author Response · Author response to Decision Letter 2]

27 Jan 2026

Dear Dr Wong and Distinguished Reviewers,

We are most grateful for your thoughtful and constructive feedback on this manuscript, and we deeply appreciate the time and expertise you invested in the review process.

Response to Reviewer #1 (Dr Weijian Huang)

Dear Dr Huang,

Thank you once again for your thoughtful and constructive feedback on our manuscript and for noting that the previous revisions had addressed your original concerns and improved the overall quality of the paper.

In your most recent comment, you kindly suggested reintroducing a clear conclusion signpost, for example by rewording the “Discussion” heading to “Discussion and Conclusion” and integrating the “Limitations and future directions” section into this closing part of the paper.

We fully agree with the importance of providing readers with an explicit structural cue for the conclusion.

In revising the manuscript, we chose to address this concern by structuring (i.e., re-arranging text without writing new text) the final part of the paper into a “Discussion” section (with the subheadings “Implications for theory” and “Implications for practice”) followed by a separate “Conclusion” section as a main heading. In the Discussion, we synthesise the key findings and elaborate the theoretical and practical implications of the study, whereas the standalone Conclusion section integrates the limitations and outlines directions for future research, rather than presenting these under an additional, separate heading. Our intention was to preserve a clear, conventional end point to the paper that functions as a strong signpost, while ensuring that both implications and future directions are tightly connected to the overall argument of the manuscript.

We hope that this structure meets the spirit of your suggestion, namely to avoid losing an explicit conclusion heading and to integrate limitations and future directions into the concluding part of the manuscript. We are very grateful for your detailed and insightful comments throughout the review process, which have significantly strengthened the clarity and contribution of the paper.

Yours sincerely,

Christian Stamov Roßnagel.

Response to Reviewer #2

Dear Reviewer 2,

Thank you very much for your thorough review and for your encouraging feedback on our manuscript. We are delighted that you consider the paper well-suited for publication at this stage and appreciate your recognition of the revisions we have undertaken.

We also thank you for your thoughtful remark regarding the sample size. As you note, the study is based on a modest sample, and we have therefore been careful to present the findings with appropriate caution, explicitly acknowledging the limitations for statistical power and generalisability, and outlining avenues for future research to address these constraints. We are grateful that you view this treatment of the limitation and future research directions as clear and satisfactory.

Finally, we are very pleased that you enjoyed the cross-cultural perspective offered by the paper. One of our central aims was to contribute to a more nuanced understanding of how constructive alignment operates in international and culturally diverse higher education contexts, and your positive evaluation is very encouraging in this regard.

Thank you again for your careful engagement with our work and for your supportive comments.

Yours sincerely,

Christian Stamov Roßnagel.

---

## [Decision Letter · Decision Letter 2]

15 Feb 2026

In transit: cross-cultural differences in international students’ constructively aligned learning experience

PONE-D-25-45434R2

Dear Dr. Christian,

We’re pleased to inform you that your manuscript has been judged scientifically suitable for publication and will be formally accepted for publication once it meets all outstanding technical requirements.

Kind regards,

Wei Lun Wong

Academic Editor

PLOS One

Additional Editor Comments (optional):

Reviewers' comments:

Reviewer's Responses to Questions

**Comments to the Author**

Reviewer #1: All comments have been addressed

2. Is the manuscript technically sound, and do the data support the conclusions?

Reviewer #1: Yes

3. Has the statistical analysis been performed appropriately and rigorously?

Reviewer #1: Yes

4. Have the authors made all data underlying the findings in their manuscript fully available?

Reviewer #1: Yes

5. Is the manuscript presented in an intelligible fashion and written in standard English?

Reviewer #1: Yes

Reviewer #1: Thanks once again for your revision. I think the paper meets the criteria of the journal, so I recommend acceptance.

**Do you want your identity to be public for this peer review?** For information about this choice, including consent withdrawal, please see our For information about this choice, including consent withdrawal, please see our Privacy Policy .

Reviewer #1: **Yes:** Weijian HuangWeijian Huang

---

## [Editor Report · Acceptance letter]

PONE-D-25-45434R2

PLOS One

Dear Dr. Stamov Roßnagel,

I'm pleased to inform you that your manuscript has been deemed suitable for publication in PLOS One. Congratulations! Your manuscript is now being handed over to our production team.

Kind regards,

on behalf of

Dr. Wei Lun Wong

Academic Editor

PLOS One